# Structure-Based Pharmacophore Modeling, Virtual Screening, Molecular Docking, ADMET, and Molecular Dynamics (MD) Simulation of Potential Inhibitors of PD-L1 from the Library of Marine Natural Products

**DOI:** 10.3390/md20010029

**Published:** 2021-12-25

**Authors:** Lianxiang Luo, Ai Zhong, Qu Wang, Tongyu Zheng

**Affiliations:** 1The Marine Biomedical Research Institute, Guangdong Medical University, Zhanjiang 524023, China; 2The Marine Biomedical Research Institute of Guangdong Zhanjiang, Zhanjiang 524023, China; 3Southern Marine Science and Engineering Guangdong Laboratory (Zhanjiang), Zhanjiang 524023, China; 4The First Clinical College, Guangdong Medical University, Zhanjiang 524023, China; zhongai@gdmu.edu.cn (A.Z.); wang15728276376@hotmail.com (Q.W.); dzheng@gdmu.edu.com (T.Z.)

**Keywords:** PD-L1, virtual screening, pharmacophore modeling, ADME, molecular dynamics

## Abstract

Background: In the past decade, several antibodies directed against the PD-1/PD-L1 interaction have been approved. However, therapeutic antibodies also exhibit some shortcomings. Using small molecules to regulate the PD-1/PD-L1 pathway may be another way to mobilize the immune system to fight cancer. Method: 52,765 marine natural products were screened against PD-L1(PDBID: 6R3K). To identify natural compounds, a structure-based pharmacophore model was generated, following by virtual screening and molecular docking. Then, the absorption, distribution, metabolism, and excretion (ADME) test was carried out to select the most suitable compounds. Finally, molecular dynamics simulation was also performed to validate the binding property of the top compound. Results: Initially, 12 small marine molecules were screened based on the pharmacophore model. Then, two compounds were selected for further evaluation based on the molecular docking scores. After ADME and toxicity studies, molecule 51320 was selected for further verification. By molecular dynamics analysis, molecule 51320 maintains a stable conformation with the target protein, so it has the chance to become an inhibitor of PD-L1. Conclusions: Through structure-based pharmacophore modeling, virtual screening, molecular docking, ADMET approaches, and molecular dynamics (MD) simulation, the marine natural compound 51320 can be used as a small molecule inhibitor of PD-L1.

## 1. Introduction

Blocking the interaction of PD-1/PD-L1 and PD-1/PD-L1 pathway modulators has shown unprecedented clinical efficacy in a variety of tumor models [1,2,3,4]. Existing studies have shown that the programmed cell death protein 1 (PD-1)/programmed cell death ligand 1 (PD-L1) signaling pathway can induce tumor-specific T cell apoptosis by inhibiting T cell activation [5]. It plays a role in immune escape and immune suppression under pathological conditions such as inflammation [6,7,8,9,10,11].

Because PD-L1 is highly expressed in a variety of tumor cells, after PD-L1 binds to PD-1, T cell activation is inhibited, and T cells are in a state of immune tolerance [12]. At this time, the immune system cannot kill the cancer cells, and tumor immune escape occurs [13]. Therefore, this type of inhibitor has a wide range of tumors, especially for tumors with high PD-L1 expression, and the response rate is higher.

Therefore, a targeted inhibitor designed for PD-L1 can cut off the signal pathway and activate T cells [14]. Therefore, its immune tolerance is relieved, T cells are mobilized to kill tumors, and tumor treatment is realized. Anti-PD-L1 monoclonal antibody is one of the most important drugs for lung cancer immunotherapy [15]. However, therapeutic antibodies also show some disadvantages. For example, tumor penetration rate is low, it is difficult to overcome physiological barriers, and there is a lack of oral bioavailability, high manufacturing cost, inaccessibility to intracellular targets, and immune-related adverse events (irAE) [16]. Using small molecules to regulate the PD-1/PD-L1 pathway may be another way to mobilize the immune system to fight cancer. In 2015, Zak et al. reported the crystal structure of the hPD-1/hD-L1 complex, which is generally considered to provide important receptor-ligand interactions, and they are reasonable structure-based drugs on the surface of PD-L1 Design (SBDD), which provides several major active sites [17]. In recent years, several small molecule drugs that can bind to PD-L1 and inhibit the interaction of PD-1/L1 have been discovered [18,19]. Therefore, it is of great significance to generate simple, stable, and efficient PD-L1 small molecule inhibitors.

Due to its special ecological environment, the ocean contains rich natural products. With the development of terrestrial resources, the marine environment provides a new field for research. By the end of 2020, it is estimated that more than 29,000 marine natural products have been found, and marine natural products have also received increasing attention from scientists [20]. It has been found that marine natural products have different structural characteristics from terrestrial natural products with various biological activities such as antifungal, antiviral, anti-parasitic, anti-tumor, and anti-inflammatory [21,22]. Natural products are the best choice as a source of new drugs [23], and marine organisms are thus considered as the latest source of bioactive natural products related to terrestrial plants and non-marine microorganisms [24]. We have collected three marine natural product databases: Marine Natural Product Database (MNPD) [25], Seaweed Metabolite Database (SWMD) [26], and Comprehensive Marine Natural Product Database (CMNPD) [27]. In this study, 52,765 kinds of marine natural products were virtually screened by targeting PD-L1. In order to predict a variety of marine natural products that may inhibit PD-L1. We hope to provide new options for the development of new anti-tumor drugs.

Structure-based pharmacophore modeling, virtual screening, molecular docking, ADMET approaches, and molecular dynamics (MD) simulation were performed on a library of marine natural products to find the novel compounds against PD-L1, which are demonstrated as Figure 1.

## 2. Results

### 2.1. Structure-Based Pharmacophore Modeling and Virtual Screening

#### 2.1.1. Pharmacophore Model Establishment

Pharmacophore describes the three-dimensional arrangement of basic spatial and electronic characteristics to achieve the best combination of ligands and macromolecules. The main application area of the pharmacophore model is database search. By searching the compound database through the pharmacophore model, it is possible to find biologically active compounds with specific targets, and to find new chemical entities with similar biological activities and different skeleton structures. Depending on the available target or known ligand information, the design of the pharmacophore model can be structure-based or ligand-based [28,29,30]. In this study, the available PD-L1 information of its small molecule inhibitors was used to construct the structure-based pharmacophore model. Hence, the 7 pharmacophore models containing the feature set were generated (Table 1) [31,32,33,34,35]. Among the generated models, 6R3K composed of DHHHNP chemical characteristics with the highest selectivity score (16.25) was selected as the best model. 

Therefore, the structure-based pharmacophore model was constructed based on PD-L1 (6R3K) and small molecule JQT (Figure 1). Ten pharmacophore models were generated. According to careful selection, we applied ligand 8 as the pharmacophore model for screening. DS software was used to generate key chemical features based on the pharmacophore model and co-crystalized ligand JQT and pharmacophore ligand 08 in Figure 2b. The total number of pharmacophores is 8. Two of them are hydrophobic, two hydrogen bond acceptors, two hydrogen bond donors, one positively charged ion center, and one negatively charged ion center (Figure 2c).

#### 2.1.2. Pharmacophore Model Validation

Verification is necessary to obtain true pharmacophore analysis and to evaluate the quality of molecular models [36]. Before database screening, the structure-based pharmacophore model established in this study was verified to verify whether the pharmacophore has a good ability to distinguish between active and inactive molecules. Receiver operating characteristic curve (ROC curve for short) is also known as sensitivity curve. The ROC curve graph is a curve reflecting the relationship between sensitivity and specificity. The X axis of the abscissa is 1-specificity, also known as false positive rate (false positive rate), the closer the X axis is to zero, the higher the accuracy rate; the Y axis of ordinate is called sensitivity, also known as true positive rate (sensitivity), the larger the Y-axis, the better the accuracy. According to the curve position, the whole graph is divided into two parts. The area under the curve is called AUC (area under curve), which is used to indicate the accuracy of prediction. The higher the AUC value, the larger the area under the curve, indicating the prediction, the higher the accuracy rate. The closer the curve is to the upper left corner (the smaller the X, the larger the Y), the higher the prediction accuracy rate [37]. In our verification process, the AUC (area under the ROC curve) at 1% threshold is 0.819 (Figure 3), which proves that our model has ability to distinguish between truly active substances and decoy compounds.

#### 2.1.3. Virtual Screening Based on Pharmacophore

A marine natural product library containing a total of 52,765 compounds was used for virtual screening based on pharmacophores against the generated pharmacophore models. A total of 12 compounds that meet the characteristics of all pharmacophores were generated. Compounds labeled HIT were retrieved and stored for further evaluation.

### 2.2. Molecular Docking

Molecular docking is an important part of the drug design process. This study aims to evaluate the binding ability of HITS compounds to the target PD-L1 protein. According to the previously obtained binding sites, a receptor grid with X = −7.1, Y = 59.3, and Z = −19.5 was prepared.

AutoDock was used to dock a specific number of drugs with PD-L1, and evaluate its binding ability, which is in line with the characteristics of the pharmacophore model. Among them, the binding affinity of compound 37080 and compound 51320 are −6.5 kcal/mol and −6.3 kcal/mol (Table 2), and their binding affinity is better than that of the PD-L1 inhibitor used in the process of generating the main pharmacophore model (−6.2 kcal/mol). The interaction of compound 37080 in the docking complex is shown in Figure 4a,b, and the interaction of compound 51320 is shown in the Figure 4c,d. In compound 51320 with good docking performance, it can be observed that the compound forms a hydrogen bond with Ala121, the oxygen atom interacts with the residue Asp122, and there is an ionic interaction with the residue ASP122 (Figure 4d). More importantly, the Pi–Pi interaction established between residue Ile54 and the compound and the Pi–Sigma interaction between residue Tyr123 and the compound also played a key role in ligand–receptor binding. Obviously, the rich interaction types between compound 513320 and the protein allow the best docking results between them. It can be seen from the interaction analysis that the docking result is reliable and the selected compound can be further analyzed.

### 2.3. Analysis of Pharmacophore Characteristics

The pharmacophore characteristics of compounds play an important role in the molecular recognition process of targeted biological macromolecules. The pharmacophore of a compound can be described according to the characteristics of H, AR, HBA or HBD, PI, and NI. This helps to identify and design new drugs for the treatment of selected diseases.

These features retain the necessary geometric arrangement of atoms required to produce a specific biological reaction. Therefore, the characteristics of the pharmacophore we generated were analyzed. As shown in Figure 5a, the overlap of the ligand and pharmacophore characteristics shows that the selected compound should be effective for our target protein. Superimposition of the top12 hit compounds on to the pharmacophore model was shown in Figure 5b and superimposition of JQT on to the pharmacophore model was shown in Figure 5c. 

### 2.4. ADME and Toxicity Test ADME Properties Analysis

Swiss-ADME is a website (https://www.swissadme.ch, accessed on 9 October 2021) which allows the user to draw their respective ligand or drug molecule or include SMILES data from PubChem and provides parameters such as lipophilicity (iLOGP, XLOGP3, WLOGP, MLOGP, SILICOS-IT, Log P0/w), water solubility Log S (ESOL, Ali, SILICOS-IT), drug-likeness rules (Lipinski, Ghose, Veber, Egan, and Muegge), and Medicinal Chemistry (PAINS, Brenk, Lead-likeness, Synthetic accessibility) [38]. The ADME prediction study of the designed compounds demonstrated Table 3. The Swiss-ADME section gives a physicochemical property of possible oral drug candidates according to five different rules determined by Lipinski, Ghose, Veber, Egan, and Muegge [39,40,41,42]. The reference value of Log S for moderately soluble and highly soluble molecules ranged from −4 to −6 and −2 to −4, respectively. According to the results, all molecules are classified as moderately soluble and highly soluble. All these parameters infer that 51320 is close to a drug-like molecule.

### 2.5. Toxicity Analysis

In order to better select lead compounds, the measurement of toxicity within silicon is an important step before the candidate drug goes into clinical trials. Calculation-based electronic toxicity measurement is widely used because of its accuracy and accessibility. It can provide information on any synthetic or natural compound. In order to determine the toxicity and adverse effects of the two selected compounds, we used the freely available testing tool ProTox-II server [43]. The software evaluates several toxicological parameters, such as acute toxicity, liver toxicity, cytotoxicity, carcinogenicity, mutagenicity, and immunotoxicity, and is based on the predicted median lethal dose (LD50) (in mg/kg body weight) (Table 4). According to the ProTox-II server, compound 51320 belongs to 4 types of toxicity, with LD 50 of 300–2000 mg/kg, which is harmful when administered orally. Compound JQT belongs to grade 4 toxicity, and its LD 50 value is 800 mg/kg. However, it was active in cytotoxicity. Combined with the ADME results, 51320 is closer to drug-like molecules than JQT. Subsequently, MD simulations were performed on 51320 molecules and JQT.

### 2.6. Structure-Based Pharmacophore Modeling and Virtual Screening

MD simulations are used to explore the binding stability of the protein–ligand docking complex. MD simulations also provide information about molecular interactions within a reference time or provide valuable assessments in explaining drug resistance [44,45]. In this paper, MD simulation methods are used to analyze the docking file of a complex of a natural compound and PD-L1 protein to determine the stability and intermolecular interaction between the protein and the molecule within a 100 ns time interval. The trajectory of MD is extracted, and the simulation results of protein–ligand (P-L) interaction mapping based on RMSD and RMSF are described. The results show that compound 51320 is stable. 

#### 2.6.1. RMSD Analysis

In order to obtain the equilibrium time of each simulated protein ligand complex during the MD simulation, the root mean square deviation (RMSD) of the backbone was calculated. RMSD diagrams are commonly used to evaluate the time required for the system to reach structural equilibrium and estimate the duration of running simulations. RMSD is an important parameter for estimating changes or changes in molecular conformation. Due to the sudden change of structural conditions, the RMSD value of the analog complex including the reference suddenly increased, which is related to the protein crystallization method. The latter effect is expected, because in the crystal structure, the protein is rigid, and when it is solvated in the water tank, it resumes its dynamic motion.

The complex system with a time frame x should have an RMSD that can be calculated from the following Equation (1).
(1)$$RMSD_{x}=\sqrt{\frac{1}{N}\overset{N}{\underset{i=1}{\sum }}\left({r^{\prime }}_{i}\left(t_{x}\right)\right)-r_{i}\left(t_{ref}\right){)}^{2}}$$

Here, the $RMSD_{x}\;$is the calculation of RMSD for the specific number of frames, N is the number of selected atoms, $t_{ref}$ is the reference or mentioned time, and $r^{\prime }$ is the selected atom in the frame x after super imposing on the reference frame, $t_{x}$ is the recording intervals.

As shown in Figure 6a, the RMSD of selected compounds 51320 and JQT were analyzed to determine whether the system has been balanced. The system of compound 51320 was in equilibrium after 105 ns, and finally stabilized at 0.33 nm, which reflected the good stability of the whole system to some extent. In addition, the docking score of the compound was −6.3 kcal/mol. The JQT system was in equilibrium after 100 ns, but finally stabilized at 0.45 nm, and the score of the JQT complex was −6.2 kcal/mol. Overall, both systems could be in equilibrium after 150 ns simulation, and the stability of the two systems was good. Compound 51320 had better system stability. 

#### 2.6.2. RMSF Analysis

In order to determine the deviation of the ligand from the initial posture and the degree of movement of protein residues, the root mean square fluctuation (RMSF) values of all sampled conformations during the 30 ns simulation were also calculated. RMSF fluctuates greatly, indicating that the residue is unstable, otherwise the residue is stable. The RMSF for residue i  was calculated from the following Equation (2).
(2)$$RMSF_{i}=\sqrt{\frac{1}{T}}\overset{T}{\underset{t=1}{\sum }}<\left({r^{\prime }}_{i}\left(t\right)\right)-r_{i}\left(t_{ref}\right){)}^{2}>$$
where T is the overall trajectory time, $r_{i}$ is the residue location, $t_{ref}$ is the reference time, $r^{\prime }$ is the location of atoms in residue i after aligned on the reference, and the angle brackets (〈 〉) are the average of the square distance.

The RMSF of two compounds was analyzed to measure the displacement of specific atoms during the simulation. In Figure 6b, the final image results of the two systems basically overlap between 0.05 and 1.25 nm, while RMSF values of docking pocket and residue generating interaction force are both lower than 0.25 nm, which to some extent indicates that the flexibility of the two systems is low, and the overall effect of compound binding is better. 

### 2.7. MM/GBSA Analysis

Molecular mechanics Poisson–Boltzmann surface area (MM/PBSA) is an effective and reliable method for calculating the free energy of binding of small inhibitors to their protein targets. Another important indicator that considers the potential affinity of a ligand to its target is the free energy of binding calculated using MM-PBSA and MD. In general, complexes with lower binding free energy can be considered more stable, and their ligands are expected to have higher activity and potency. We summarize the interaction energy and binding free energy of the two complexes in Figure 7a,b, respectively. The MM/GBSA of the complex system is calculated from a single trajectory collected from the respective 100 ns simulation (Table 5). The analysis of the contribution of each energy term shows that the electrostatic interaction of 51320 (−179.032 kJ/mol) (Figure 8b) is much stronger than the corresponding term of JQT (Figure 8a) and PD-L1 (12.196 kJ/mol). Therefore, the compounds screened will be able to maintain a lasting interaction with the desired protein. The quantitative information on the contribution of each amino acid residue to energy is very helpful for a better understanding of the binding mechanism of inhibitor molecules. The analysis of the selected compounds in Figure 8 revealed that MET-115, TYR-56, and ILE-54 have high energy contributions in JQT and 51320. It can be seen from the above results that the selected compound can maintain long-term interaction with the binding site of the PD-L1 protein, resulting in the inhibition of the target protein.

## 3. Discussion

In recent years, anti PD-L1 monoclonal antibodies have also shown positive reactions in clinical trials of various malignant tumors. However, antibody drugs have some shortcomings, such as immunogenicity problems and poor tumor tissue permeability, resulting in a low overall response rate of PD-1/PD-L1 antibody drugs [29]. At present, small molecule inhibitors based on PD-1/PD-L1 are gradually recovering. Marine natural products are closely related to the fields of drug discovery and molecular biology, and have always attracted the attention of the scientific community. Therefore, our research aims to use marine natural products to perform virtual screening of PD-L1 detection sites.

In this study, we collected 7 structures of PD-L1 with low molecular mass inhibitor by reading the literature. A structure-based pharmacophore model was constructed by using DS 4.5, and 6R3K with the highest selection score was selected. The pharmacophore model constructed by the complex was used to screen the marine natural product database. The pharmacophore model is validated by the active compound, and the AUC under the ROC curve indicates that the model has good distinguishing ability. The validated pharmacophore model was used in the virtual screening process. A total of 12 compounds were retrieved for hits and further screened by molecular docking methods. According to the molecular docking score, the first two compounds with a better binding score than the original ligand JQT (−6.2 kcal/mol) were selected for further verification.

We conducted interaction analysis from the perspective of molecular docking. The rich interaction types between compound 513320 and protein can be seen, the docking result is reliable, and the selected conformation can be further analyzed. In addition, quantum mechanics/molecular mechanics (QM/MM) calculations can be performed on the complex to select conformations from docking simulations [46]. After years of development and calibration, QM/MM hybrid method has become an indispensable tool to study the dynamics of a variety of chemical and biochemical processes. For example, an article uses a molecular docking method combining quantum mechanics and molecular mechanics (QM/MM) to determine the resuscitation pathway of inhibited AChE [47]. Another article studied the reaction mechanism between oxime and *Mm*AChE, using the sequential QM/docking (MM) method [48]. QM/MM is mostly used to characterize and study the transition state and activation energy of enzyme reactions. The conformation calculated by this method describes the surrounding environment in more detail. However, the calculation becomes more complicated and not easy to control. We selected the conformation by analyzing the interaction between molecules and combining scores, which not only pays attention to the binding mode but also has scores for reference. However, the influence of the environment is not considered, and some compounds will change the docking conformation due to environmental changes. All in all, it is better to use QM/MM for the selection of the molecular conformation of the enzyme’s active target. It is more visual and convenient to use molecular interaction force to choose the conformation of other molecules. In the future, further QM/MM research on PD-L1 and molecules can be carried out for better selection of molecular conformations.

The two selected compounds 51320 and 37080 have been evaluated based on ADME characteristics, and the 51320 showed good ADME characteristics. Compound 37080 violates Lipinski’s Five Rules, so this compound was skipped for further evaluation. Compounds with good ADME properties were further evaluated for toxicity properties to measure harmful effects on humans or animals. Toxicity analysis found that the selected compound 51320 has very low toxicity and JQT has cytotoxicity. Our selected compound 51320 has no cytotoxicity and is better than JQT.

The MD simulation method identifies the stability of the compound to the protein binding site. The 150 ns simulation trajectory was searched and analyzed based on RMSD and RMSF to confirm the stability of the compound and protein binding site. In addition, the MM/GBSA calculated from a single trajectory found a high ΔG binding value, indicating that the selected protein-ligand complex has long-term simulation stability.

All in all, marine natural products provide a lot of information for the discovery of new drugs. Through virtual screening, small-molecule inhibitors of PD-L1 were efficiently identified from more than 52,000 marine natural products. The expansion and further optimization of the screening range can finally identify useful immunomodulators to help improve public health.

## 4. Materials and Methods

### 4.1. Structure-Based Pharmacophore Modeling and Virtual Screening

#### 4.1.1. Complex-Based Pharmacophore Modeling

Our first goal in this project was to collect as much information as possible about PD-L1 as an inhibitor target. Through a literature search, we downloaded 7 PD-L1 compounds and small molecule inhibitors from the PDB. Discovery Studio 4.5 was used to establish a pharmacophore model based on the compound (Table 1). The pharmacophore model with the highest score was used to screen the compound.

#### 4.1.2. Pharmacophore Model Validation

In the modeling process, the receiver operating characteristic (ROC) curve analysis and verification method built in DS 4.5 was used. Pharmacophore validation helps to assess the potential properties of active and inactive compounds, usually derived from specific protein-ligand interactions. A total of 90 active antagonists obtained from patent and literature searches were used to validate the pharmacophore model, which is composed of 60 active compounds and 30 inactive compounds. Then, the generated pharmacophore was verified by using the verification option in the receptor ligand pharmacophore generation protocol implemented in DS 4.5. From the area under the ROC curve (AUC), we can judge whether a pharmacophore has successfully selected active ingredients and removed inactive ingredients [49,50]. The area under the ROC curve (AUC) is 0 ≤ A ≤ 1. When A > 0.5, the closer A is to 1, the higher the diagnostic accuracy. When A = 0.5, the diagnosis does not work at all. When A < 0.5, it does not meet the actual situation.

#### 4.1.3. Virtual Screening Based on Pharmacophore

The marine small molecule databases (MND, SWMD, CMNPD) were screened according to the characteristics of the pharmacophore. DS 4.5 created and acquired 3D models in the case of protein-ligand interactions. These compounds were directly transferred to the database list for rapid virtual screening based on pharmacophore characteristics. According to the pharmacophore matching score, the fitted hit compounds were ranked and further verified. Of the 20 molecules obtained, a total of 12 hits were selected after careful visual inspection.

### 4.2. ADME and Toxicity Test

#### 4.2.1. ADME

ADME is important to analyze the pharmacodynamics of the proposed molecule which could be used as a drug. The Swiss ADME server (http://www.swissadme.ch/, accessed on 9 October 2021) was used to evaluate the selected ligands which were harvested from PubChem, which was done on the basis canonical SMILES [38]. The ADME properties of the chosen compounds were calculated. The major ADME-associated parameters such as pharmacokinetic properties and the solubility of the drug were considered. The values of the observed properties are presented in Table 3.

#### 4.2.2. Toxicity Test

Calculation-based methods have made it possible to obtain a safety profile of the desired compound to measure toxicity through computer methods. ProTox-II (http://tox-new.charite.de/protox_II/, accessed on 9 October 2021) server was used to determine the toxic effects of the two selected compounds [51]. The ProTox-II server predicts the median lethal dose (LD50), organ toxicity (hepatotoxicity), and toxicological endpoint (immunotoxicity and cytotoxicity) of the query molecule.

### 4.3. MD Simulation

In order to further verify the results obtained, the second docking program CDOCKER on DS 4.5 was adopted. The results were evaluated based on the interaction energy of CDOCKER, and a higher interaction energy of CDOCKER meant greater beneficial binding.

After docking, the compound with the highest binding energy for each target is simulated by MD simulation to check the stability of the compound in the binding pocket. Then, GROMACS 2019.1 software package [52], gromos54a7atb.ff force field and single point charge (SPC216) model was used for molecular dynamics simulation of the 150 ns. In order to ensure the total charge neutrality of the simulated system, a corresponding number of sodium ions was added to replace the water molecules in the three systems to produce a solvent box of appropriate size. Then, periodic boundary conditions (PBC) [53] were applied in the three directions of the system. Using the gromos54a7_atb force field, the force field parameters of the entire atom can be obtained from the ATB website (http://atb.uq.edu.au/, accessed on 9 October 2021). A first pass (EM) was conducted to minimize the energy of 50,000 steps of the entire system at 300 K, then through MD simulation with location constraints, through NVT collection (constant particle number, volume and temperature), and finally through NPT collection (constant particle number, pressure, and temperature) [53]. In addition, we balanced enzymes, ligand molecules, and solvents. 

### 4.4. MM/GBSA

Improved MM-PBSA or Molecular Mechanics–Poisson Boltzmann Surface Area is an opensource software used to calculate the free energy of binding between the receptor and the inhibitor. As a scoring function, MM-PBSA has been used in the calculation method of drug design [54]. In this study, MM-PBSA was used to determine the binding free energy of JQT and molecule 51320, respectively.

The following Equation (3) describes the binding free energy:(3)$$G_{binding}=G_{complex}-\left(G_{protein}+G_{ligand}\right)$$

The free energy of protein-inhibitor complex is represented by *G_Complex_*, the free energy of protein in solvent is represented by *G_protein_*, and the free energy of inhibitor in solvent is represented by *G_ligand_*.

## 5. Conclusions

PD-L1 has become a therapeutic target for many malignant tumors. In this study, a structure-based pharmacophore model was generated using the crystal structure of PD-L1 (6R3K) and the combined small molecule inhibitor JQT. These were used for virtual screening of a marine natural product database. Molecular docking, ADME analysis, and toxicity studies were performed on the obtained compounds. Subsequently, the molecule 51320 was selected for molecular dynamics simulation and MM/GBSA methods, revealing that 51320 is a potential small molecule inhibitor that helps inhibit PD-L1. The small molecule can be further evaluated through different laboratory-based experimental techniques to help determine the activity of the compound, thereby providing an alternative to immunotherapy.

## Figures and Tables

**Figure 1 marinedrugs-20-00029-f001:**
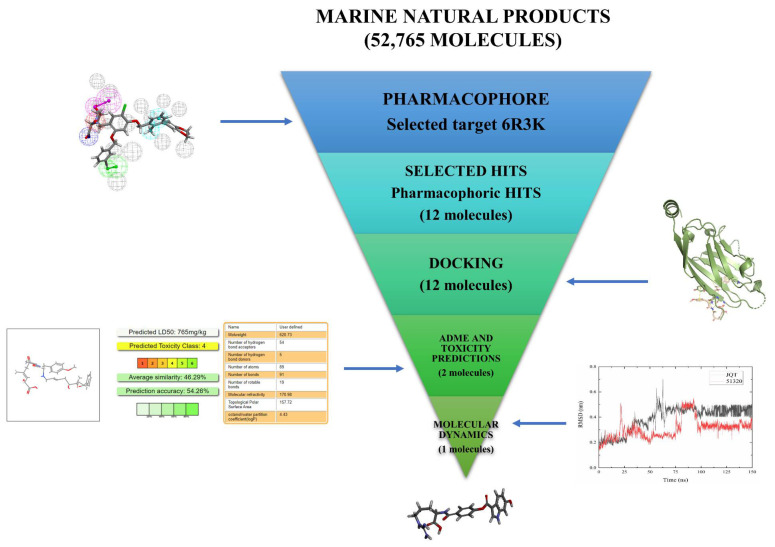
The virtual screening workflow (VSW) used in this work for the identification of hit molecules targeting PD-L1. A workflow overview of pharmacophore modeling, virtual screening, molecular docking, absorption, distribution, metabolism, elimination, and toxicity (ADMET) approaches, and MD simulation.

**Figure 2 marinedrugs-20-00029-f002:**
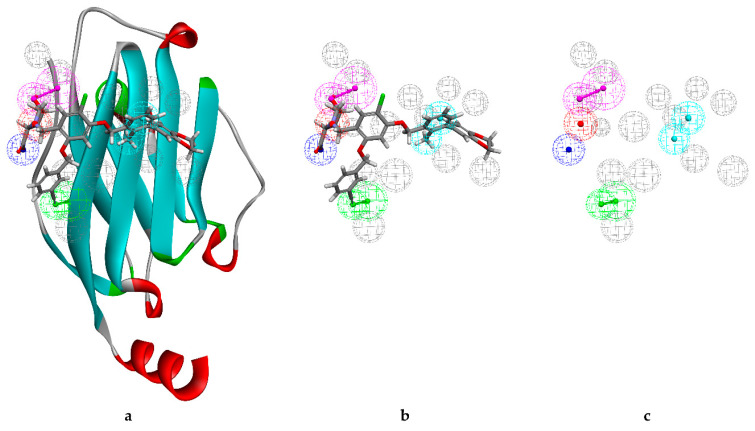
The overall pharmacophore model generated during the study. (**a**) PD-L1 crystal (PDB ID:6R3K) with the co-crystalized ligand JQT and pharmacophore interaction map; (**b**) co-crystalized ligand JQT and pharmacophore ligand 08; (**c**) pharmacophore ligand 08.

**Figure 3 marinedrugs-20-00029-f003:**
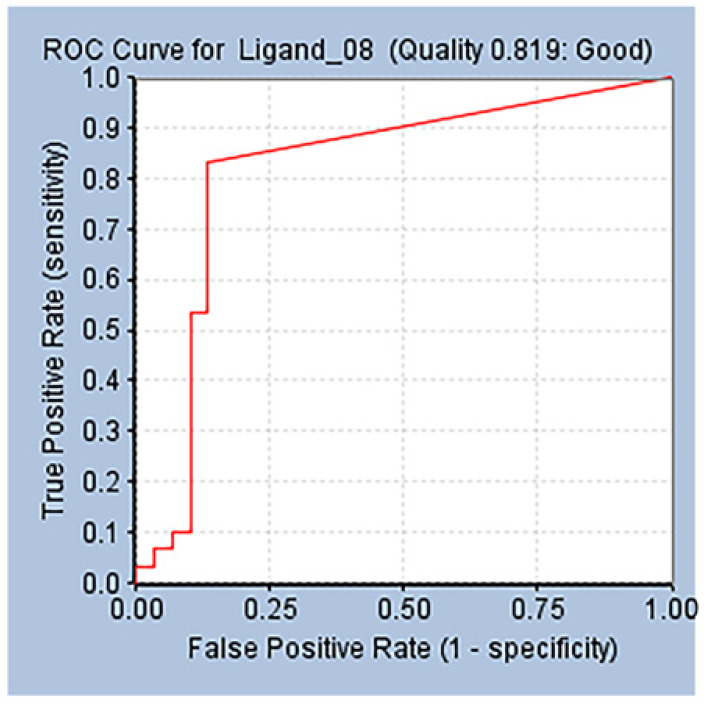
Receiver operating characteristic (ROC) curve of Ligand_08.

**Figure 4 marinedrugs-20-00029-f004:**
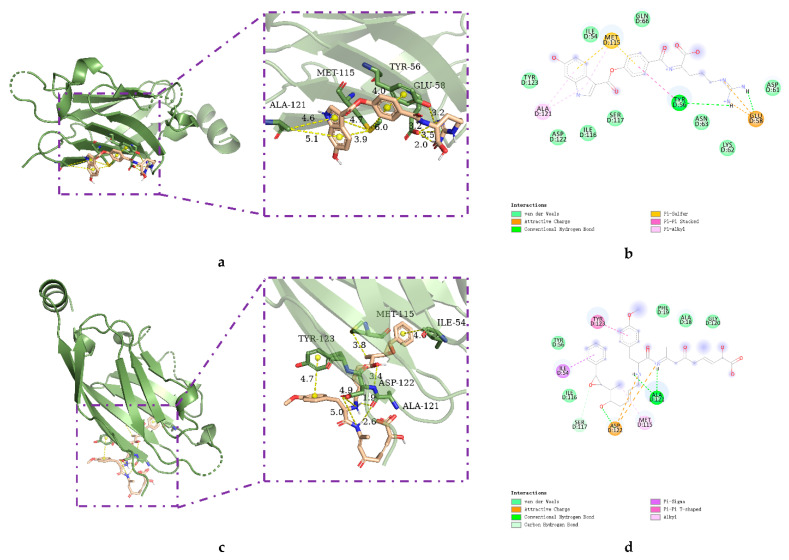
Interaction between the protein–ligand complex. (**a**) Three-dimensional binding mode of the 37080 and protein complex; (**b**) two-dimensional binding mode of the 37080 and protein complex; (**c**) three-dimensional binding mode of the 51320 and protein complex; (**d**) two-dimensional binding mode of the 51320 and protein complex.

**Figure 5 marinedrugs-20-00029-f005:**
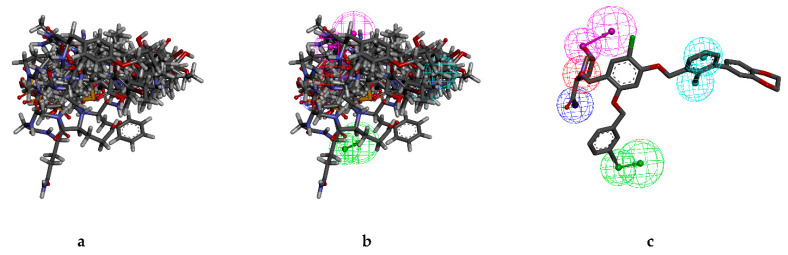
Structure-based pharmacophore. (**a**) Aligned ligands. (**b**) Selected ligands superimposed to the simplified pharmacophore model. (**c**) Ligand JQT superimposed to the simplified pharmacophore model.

**Figure 6 marinedrugs-20-00029-f006:**
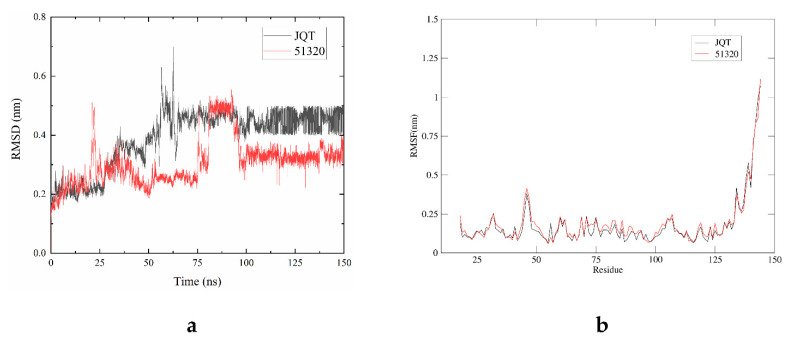
Root mean square deviation (RMSD) and root mean square fluctuation (RMSF) plots of JQT complex (black), 51320 (red). (**a**) RMSD values extracted from protein fit ligand of the protein–ligand docked complexes. RMSD plot of JQT complex (black), 51320 (red); (**b**) the RMSF graph of all complexes along with protein during 100 ns MD simulation. RMSF plot of JQT complex (black), 51320 (red).

**Figure 7 marinedrugs-20-00029-f007:**
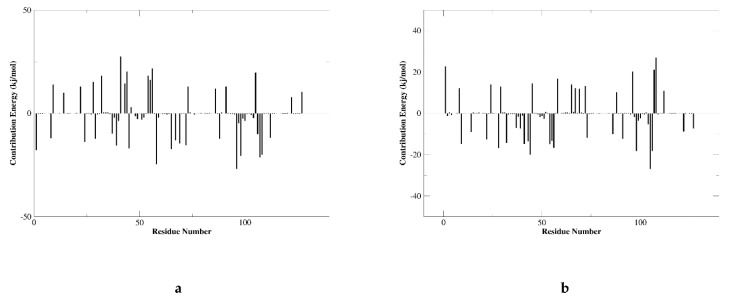
Residue wise decomposition of binding free energies obtained from the MMPBSA analyses. (**a**) JQT; (**b**) 51320.

**Figure 8 marinedrugs-20-00029-f008:**
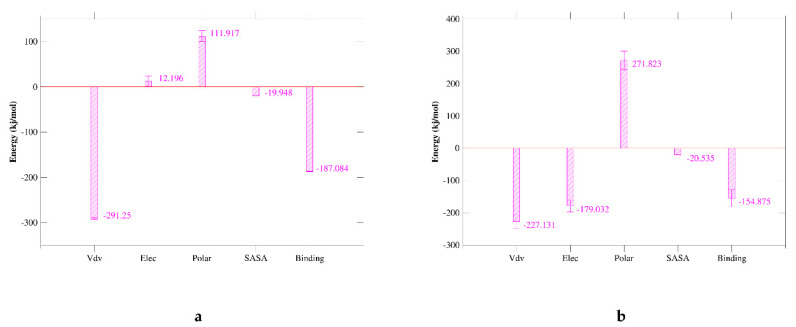
Binding energy of binding for the protein complexed with ligands JQT, 51320. (**a**) JQT; (**b**) 51320.

**Table 1 marinedrugs-20-00029-t001:** Ranking of pharmacophore models.

Compound	Number of Features	Feature Set	Selectivity Score	References
6R3K	6	DHHHNP	16.25	[31]
5NIU	6	DDHHHP	15.635	[32]
5N2F	6	AAHHNP	12.936	[33]
5N2D	6	AHHHPR	12.848	[33]
5J89	5	HHHHP	11.196	[17]
6NM8	6	HHHHHP	10.996	[34]
5J8O	6	HHHHHR	9.2594	[35]

**Table 2 marinedrugs-20-00029-t002:** Molecular docking results of JQT and 12 selected ligands from the library of marine natural products.

Molecules	2D Structure	Binding Affinity (kcal/mol)	Formula
37080	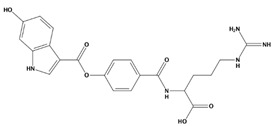	−6.5	C_22_H_23_N_5_O_6_
51320	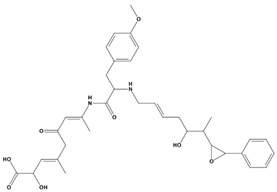	−6.3	C_35_H_44_N_2_O_8_
37113	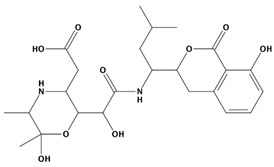	−6.2	C_24_H_34_N_2_O_9_
38010	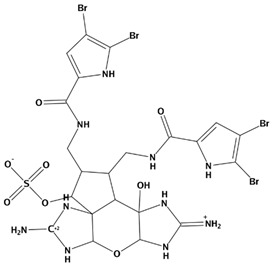	−5.7	C_22_H_25_Br_4_N_10_O_8_S
41160	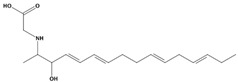	−5.2	C_18_H_29_NO_3_
32979	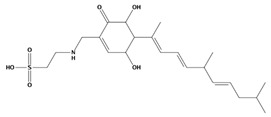	−5.1	C_22_H_35_NO_6_S
35432	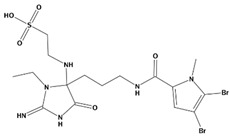	−4.7	C_16_H_24_Br_2_N_6_O_5_S
21793	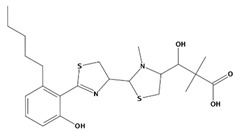	−4.6	C_23_H_34_N_2_O_4_S_2_
23671	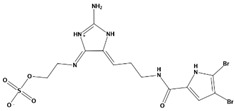	−4.6	C_13_H_16_Br_2_N_6_O_5_S
41159	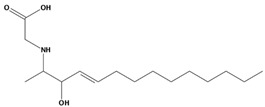	−4.5	C_16_H_31_NO_3_
35433	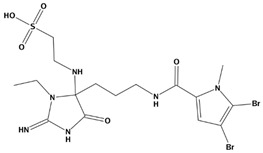	−4.4	C_16_H_24_Br_2_N_6_O_5_S
50094	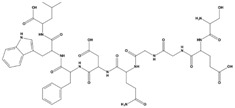	−3.9	C_47_H_63_N_11_O_16_
JQT	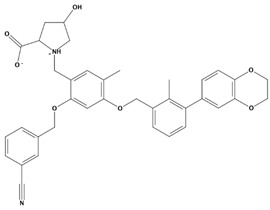	−6.2	C_36_H_33_ClN_2_O_7_

**Table 3 marinedrugs-20-00029-t003:** ADME properties of JQT and selected ligands from the library of marine natural products.

Molecule	MW	Rotatable Bonds	H-bond Acceptors	H-bond Donors	ESOL Log S	TPSA	WLOGP	GI Absorption	log Kp (cm/s)
37080	453.45	12	7	7	−3.18	190.62	1.54	0.78	Low
51320	620.73	19	9	5	−3.82	157.72	3.33	0.99	Low
37113	494.53	9	10	6	−1.91	174.65	−0.09	0.04	Low
38010	909.18	10	8	11	−5.32	271.15	−2.38	0.43	Low
41160	307.43	12	4	3	−1.6	69.56	3.22	0.35	High
32979	441.58	11	7	4	−2.04	132.31	3.13	1.28	High
35432	572.27	11	7	5	−2	165	0.94	0.31	Low
21793	466.66	9	6	3	−3.74	143.96	3.07	2.18	Low
23671	528.18	9	6	5	−3.99	166.6	1.06	−0.33	Low
41159	285.42	13	4	3	−2.01	69.56	3.11	0.14	High
35433	572.27	11	7	5	−2	165	0.94	0.31	Low
50094	1038.07	40	17	15	−0.68	449.83	−3.85	−4.93	Low
JQT	641.11	10	9	2	−5.81	121.48	5.30	2.64	Low

**Table 4 marinedrugs-20-00029-t004:** List of toxicity properties (organ toxicity, toxicity endpoints, Tox21-nuclear receptor signaling pathways, Tox21-stress response pathway) of the selected 2 compounds.

Endpoint	Target	JQT	51320
Organ toxicity	Hepatotoxicity	Inactive	Inactive
Toxicity end points	Carcinogenicity	Inactive	Inactive
	Immunotoxicity	Inactive	Inactive
	Mutagenicity	Inactive	Inactive
	Cytotoxicity	Active	Inactive
	LD50 (mg/kg)	800	765
	Toxicity class	4	4
Tox21-nuclear receptor signaling pathways	Aryl hydrocarbon receptor (AhR)	Inactive	Inactive
	Androgen receptor (AR)	Inactive	Inactive
Tox21-stress response pathways	Heat shock factor response element (HSE)	Inactive	Inactive

**Table 5 marinedrugs-20-00029-t005:** Binding energy of binding for the protein complexed with ligands JQT, 51320.

Criteria	JQT	51320
Van der Waal energy (kJ/mol)	−291.250 ± 1.797	−227.131 ± 21.896
Electrostatic energy (kJ/mol)	12.196 ± 11.229	−179.032 ± 18.056
Polar solvation energy (kJ/mol)	111.917 ± 12.103	271.823 ± 28.866
SASA energy (kJ/mol)	−19.948 ± 0.176	−20.535 ± 0.727
Binding energy (kJ/mol)	−187.084 ± 0.748	−154.875 ± 25.470

## Data Availability

The data used to support the findings of this study are included within the article.

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
