# Peer review of "Structure-Based Pharmacophore Modeling, Virtual Screening, Molecular Docking, ADMET, and Molecular Dynamics (MD) Simulation of Potential Inhibitors of PD-L1 from the Library of Marine Natural Products"

_marinedrugs, 2021, doi:10.3390/md20010029_

Round 1

Reviewer 1 Report

Whereas the work seems to be carefully done, some few points need attention before publication.  

  1. The authors could perform MD simulations for all selected compounds for evaluating the toxicity properties.

  1. A theoretical strategy able to probe the conformational profile of ligands in the enzyme active site is very important. It is well-known that, from a theoretical standpoint, molecular dynamics simulations can be used to evaluate the molecular flexibility of ligands and receptors; however, it is worth mentioning that some conformational changes occur in the time scale of only dozens of nanoseconds, which could compromise the MD simulation viability for virtual screening studies, for instance. In this regard, a theoretical strategy is to select promising configurations from the docking study is crucial to determine the theoretical accuracy. In this line, I would like to suggest to authors to include a discussion about theoretical methods for selecting conformations from docking simulations, introducing the references:

  BMC PHARMACOLOGY & TOXICOLOGY   Vol.: 19,   8  (2018) 

 LETTERS IN DRUG DESIGN & DISCOVERY   Vol. 13, 360-371 (2016) 

Author Response

Reviewer #1(Remaeks to the Author):

Whereas the work seems to be carefully done, some few points need attention before publication. 

The authors could perform MD simulations for all selected compounds for evaluating the toxicity properties.

Response: We appreciate the reviewers’ comments and suggestions. Although we agree that this is an important consideration, it is not suitable for inclusion in this manuscript, because molecular dynamics is used to observe the conformational space, equilibrium properties, and dynamic properties of molecules, not to evaluate the toxic properties of molecules. In addition, we could evaluate the toxicity of all compounds through the ProTox-II (http://tox.chari te.de/ProTox-II) server.

A theoretical strategy able to probe the conformational profile of ligands in the enzyme active site is very important. It is well-known that, from a theoretical standpoint, molecular dynamics simulations can be used to evaluate the molecular flexibility of ligands and receptors; however, it is worth mentioning that some conformational changes occur in the time scale of only dozens of nanoseconds, which could compromise the MD simulation viability for virtual screening studies, for instance. In this regard, a theoretical strategy is to select promising configurations from the docking study is crucial to determine the theoretical accuracy. In this line, I would like to suggest to authors to include a discussion about theoretical methods for selecting conformations from docking simulations, introducing the references:

  BMC PHARMACOLOGY & TOXICOLOGY   Vol.: 19, 8 (2018)

 LETTERS IN DRUG DESIGN & DISCOVERY   Vol. 13, 360-371 (2016)

Response: Thanks for your kind comments and suggestions. We agree with the reviewers’ point in the importance of the theoretical strategy able to probe the conformational profile of ligands in the enzyme active site, so we conducted an interaction analysis from the perspective of molecular docking. In compound 51320 with good docking performance, it can be observed that the compound forms a hydrogen bond with Ala121, the oxygen atom interacts with the residue Asp122, and there is an ionic interaction with the residue ASP122. More importantly, the Pi-Pi interaction established between residue Ile54 and the compound and the Pi-Sigma interaction between residue Tyr123 and the compound also play a key role in ligand-receptor binding. Obviously, the rich interaction types between compound 513320 and the protein allow the best docking results between them. It can be seen from the interaction analysis that the docking result is reliable and the selected compound can be further analyzed. In addition, quantum mechanics/molecular mechanics (QM/MM) calculations can be performed on the complex to select conformations from docking simulations [1]. After years of development and calibration, QM/MM hybrid method has become an indispensable tool to study the dynamics of a variety of chemical and biochemical processes. For example, an article uses a molecular docking method combining quantum mechanics and molecular mechanics (QM/MM) to determine the resuscitation pathway of inhibited AChE [2]. Another article studied the reaction mechanism between oxime and MmAChE, using the sequential QM/docking (MM) method [3]. We have added the methods of screening conformations in the discussion section (Line 388-409).

  1. Ryde, U. QM/MM Calculations on Proteins. Methods Enzymol 2016, 577, 119-158, doi:10.1016/bs.mie.2016.05.014.
  2. Kuca, K.; Musilek, K.; Jun, D.; Zdarova-Karasova, J.; Nepovimova, E.; Soukup, O.; Hrabinova, M.; Mikler, J.; Franca, T.C.C.; Da Cunha, E.F.F.; et al. A newly developed oxime K203 is the most effective reactivator of tabun-inhibited acetylcholinesterase. BMC Pharmacol Toxicol 2018, 19, 8, doi:10.1186/s40360-018-0196-3.
  3. Ramalho, T.C.; Cunha, E.F.F.D.; Castro, A.A.D.; Pereira, A.F.; Lima, W.E.A.D.J.L.i.D.D.; Discovery. Flexibility in the Molecular Design of Acetylcholinesterase Reactivators: Probing Representative Conformations by Chemometric Techniques and Docking/QM Calculations. 2016, 13, -.

Reviewer 2 Report

In the manuscript (ID: marinedrugs-1521082), the authors have used different computational methods such as: Structure Based Pharmacophore Modeling, Virtual Screening, Molecular Docking,  ADMET Approaches and Molecular Dynamics (MD) to find the novel compounds against PD-L1.

The manuscript is not clearly presented. The methods were described improperly, without details that sometimes can be found in the results sections e.g. molecular dynamics  length. There aren’t any references to software used, confusion which software was used and when. The method section and afterwards the results section should be described more precisely.

The discussion regarding amino acids contribution and their importance (mutagenesis data) to the ligand binding should be carried out. The molecular dynamics results should be verified (longer simulation up to 150 ns), because RMSD of compound 51320 fluctuates at the end of dynamics.

I have a few concerns that can be easily addressed:

1) A few more references should be made in the introduction so that the reader can easily find information on the effects of PL-D1 and its inhibitors in the treatment of cancer.

2) In Table 1, the references to each of the structure should be revealed.

3) Any abbreviations of names should be given e.g. DS.

4) How were chosen decoys for analysis?

5)  What mean the value of toxicity class? What does it mean that compound was active in the cytotoxicity?

Author Response

Reviewer #2(Remaeks to the Author):

In the manuscript (ID: marinedrugs-1521082), the authors have used different computational methods such as: Structure Based Pharmacophore Modeling, Virtual Screening, Molecular Docking, ADMET Approaches and Molecular Dynamics (MD) to find the novel compounds against PD-L1.

The manuscript is not clearly presented. The methods were described improperly, without details that sometimes can be found in the results sections e.g. molecular dynamics length. There aren’t any references to software used, confusion which software was used and when. The method section and afterwards the results section should be described more precisely.

Response: Thank you for the constructive comments and suggestions. In the method part, we have marked the version and model of the software, time, etc. And we described the details and revised in the result part. More detailed content builds the yellow key part of the manuscript.

The discussion regarding amino acids contribution and their importance (mutagenesis data) to the ligand binding should be carried out. The molecular dynamics results should be verified (longer simulation up to 150 ns), because RMSD of compound 51320 fluctuates at the end of dynamics.

Response: Thanks for your kind suggestions, the reviewer brings to light many salient points. To address the first point, we have added the version of the software used and the corresponding parameter references and the length of the molecular dynamics. As for the results, we have also replaced Figure 6. Root mean square deviation (RMSD) and root mean square fluctuation (RMSF) plots of JQT complex (black), 51320 (red), as shown below. More details have been included in the revised manuscript (Line 314-321, 332-337).

Figure 6. Root mean square deviation (RMSD) and root mean square fluctuation (RMSF) plots of JQT complex (black), 51320 (red). (a) RMSD values extracted from protein fit ligand of the protein–ligand docked complexes. RMSD plot of JQT complex (black), 51320 (red); (b) The RMSF graph of all complexes along with protein during 100 ns MD Simulation. RMSF plot of JQT complex (black), 51320 (red).

1) A few more references should be made in the introduction so that the reader can easily find information on the effects of PL-D1 and its inhibitors in the treatment of cancer.

Response:Thank you for the constructive comments. More details have been included in the revised manuscript (Line 53-58).

2) In Table 1, the references to each of the structure should be revealed.

Response:Thank you for the constructive comments. We have updated the table 1. Ranking of pharmacophore models, as shown below.

Table 1. Ranking of pharmacophore models.

Compound

Number of Features

Feature Set

Selectivity Score

References

6R3K

6

DHHHNP

16.25

[1]

5NIU

6

DDHHHP

15.635

[2]

5N2F

6

AAHHNP

12.936

[3]

5N2D

6

AHHHPR

12.848

[3]

5J89

5

HHHHP

11.196

[4]

6NM8

6

HHHHHP

10.996

[5]

5J8O

6

HHHHHR

9.2594

[6]

3) Any abbreviations of names should be given e.g. DS.

Response: Thanks for your constructive suggestion, which is highly appreciated. These parts were revised and highlighted in the manuscript.

4) How were chosen decoys for analysis?

Response:Thank you for your suggestions. We obtained the PD-L1small molecule inhibitors and their IC50 through patent collection and extensive text search. The 90 compounds taken for training set were classified into 60 active (IC50 <= 100nM) and 30 inactive molecules (IC50 >= 10 nM).

5)  What mean the value of toxicity class? What does it mean that compound was active in the cytotoxicity?

Response: Thank you for your suggestion. The ProTox-II server categorizes the molecules according to the Globally Harmonized System of Classification and Labelling of Chemicals (GHS). Class I is fatal if swallowed (LD50 < 5 mg/kg), Class II is fatal if swallowed (LD50 < 50 mg/Kg), Class III toxic if swallowed (LD50 < 300 mg/Kg), Class IV harmful if swallowed (LD50 < 2000 mg/Kg), Class V may be harmful if swallowed (LD50 < 5000 mg/Kg), Class VI non-toxic (LD50 > 5000 mg/Kg)[7]. Active in the cytotoxicity means potential toxicity to cells. Contact with cytotoxic substances can cause permanent cell damage and even death.

  1. Muszak, D., et al., Terphenyl-based Small-Molecule Inhibitors of Programmed Cell Death-1/Programmed Death-Ligand 1 ProteinProtein Interaction. 2021. XXXX(XXX).
  2. Skalniak, L., et al., Small-molecule inhibitors of PD-1/PD-L1 immune checkpoint alleviate the PD-L1-induced exhaustion of T-cells. 2017. 8(42).
  3. Guzik, K., et al., Small-Molecule Inhibitors of the Programmed Cell Death-1/Programmed Death-Ligand 1 (PD-1/PD-L1) Interaction via Transiently Induced Protein States and Dimerization of PD-L1. 2017: p. 5857.
  4. Zak, K.M., et al., Structural basis for small molecule targeting of the programmed death ligand 1 (PD-L1). 2016. 7(21).
  5. Perry, E., et al., Fragment-based screening of programmed death ligand 1 (PD-L1). Bioorg Med Chem Lett, 2019. 29(6): p. 786-790.
  6. Amaral, M., et al., Protein conformational flexibility modulates kinetics and thermodynamics of drug binding. 2017. 8(1): p. 2276.
  7. Drwal, M.N., et al., ProTox: a web server for the in silico prediction of rodent oral toxicity. 2014(W1): p. W53.
